# Multilevel Pain Assessment with Functional Near-Infrared Spectroscopy: Evaluating $\Delta HBO_2$ and $\Delta HHB$ Measures for Comprehensive Analysis

Muhammad Umar Khan, Maryam Sousani, Niraj Hirachan, Calvin Joseph, Maryam Ghahramani, Girija Chetty, Roland Goecke and Raul Fernandez-Rojas *

Human-Centred Technology Research Centre, Faculty of Science and Technology, University of Canberra, Canberra, ACT 2617, Australia
* Correspondence: raul.fernandezrojas@canberra.edu.au

**Abstract:** Assessing pain in non-verbal patients is challenging, often depending on clinical judgment which can be unreliable due to fluctuations in vital signs caused by underlying medical conditions. To date, there is a notable absence of objective diagnostic tests to aid healthcare practitioners in pain assessment, especially affecting critically-ill or advanced dementia patients. Neurophysiological information, i.e., functional near-infrared spectroscopy (fNIRS) or electroencephalogram (EEG), unveils the brain's active regions and patterns, revealing the neural mechanisms behind the experience and processing of pain. This study focuses on assessing pain via the analysis of fNIRS signals combined with machine learning, utilising multiple fNIRS measures including oxygenated haemoglobin($\Delta HBO_2$) and deoxygenated haemoglobin ($\Delta HHB$). Initially, a channel selection process filters out highly contaminated channels with high-frequency and high-amplitude artifacts from the 24-channel fNIRS data. The remaining channels are then preprocessed by applying a low-pass filter and common average referencing to remove cardio-respiratory artifacts and common gain noise, respectively. Subsequently, the preprocessed channels are averaged to create a single time series vector for both $\Delta HBO_2$ and $\Delta HHB$ measures. From each measure, ten statistical features are extracted and fusion occurs at the feature level, resulting in a fused feature vector. The most relevant features, selected using the Minimum Redundancy Maximum Relevance method, are passed to a Support Vector Machines classifier. Using leave-one-subject-out cross validation, the system achieved an accuracy of 68.51% ± 9.02% in a multi-class task (No Pain, Low Pain, and High Pain) using a fusion of $\Delta HBO_2$ and $\Delta HHB$. These two measures collectively demonstrated superior performance compared to when they were used independently. This study contributes to the pursuit of an objective pain assessment and proposes a potential biomarker for human pain using fNIRS.

**Keywords:** pain assessment; fNIRS; statistical features; SVM; machine learning

## 1. Introduction

Pain, despite its unpleasantness, acts as an essential biomarker in our bodies, alerting us to potential health issues, injuries, or emotional stress. Pain can be localised to a particular region, like an injury, but it can also be more widespread, as seen in many illnesses [1]. Pain is a significant issue in society as it poses a substantial public health challenge, impacts the quality of life of sufferers, and places a burden on the economy [2,3]. The economic impacts of pain are drastic, imposing a financial burden exceeding AUD 73 billion dollars annually, including AUD 48.3 billion dollars in lost productivity in Australia alone [4,5]. Furthermore, it impacts the day-to-day routines and significantly diminishes the overall quality of life. For instance, low back pain is the leading cause of disability in the world, with over 600 million people living with pain [6]. Therefore, the assessment and management of pain is essential for a wide range of clinical disorders and treatments, and its early

diagnosis plays a vital role in mitigating the risk of its progression into chronic conditions or contributing to depression or anxiety [7].

Pain is a subjective experience and its measurement is difficult. In clinical practice, two primary subjective methods are used for pain assessment: self-reports and clinical judgment [8]. The commonly accepted method to assess pain is self-report. Self-reporting techniques aim to gauge a patient's pain using verbal or numerical self-assessment tools, including methods such as visual analogue scales, verbal descriptor scales, numerical rating scales, or the McGill Pain Questionnaire [9,10]. When self-reports are not accessible or may be unreliable, clinical observations can serve as a supplementary or alternative method. Clinical judgment for pain assessment relies on examining and understanding the nature, intensity, and context of the patient's pain experience based on observations [7]. Despite their convenience and utility, subjective reports come with various limitations such as inconsistent measurement scales and variations in how pain is understood by medical professionals and patients. Furthermore, these methods cannot be effectively employed in cases involving children or patients with neurological disorders.

In order to address these limitations, researchers have turned to the analysis of the neurological aspects of pain using objective methods such as neuroimaging [11]. For instance, Wager et al. [12] developed a system that employs machine learning to analyse data obtained from functional magnetic resonance imaging (fMRI). Their work demonstrated the potential to identify a consistent neurological signature of pain at the individual level. While fMRI-based objective assessments of pain have made significant progress in understanding the brain's pain mechanisms, the size and cost of MRI scanners and other conventional neuroimaging tools (such as positron emission tomography) make them impractical for routine clinical use [13]. This limitation has increased the interest in portable neuroimaging devices that offer similar technical advantages to fMRI. One such technology is functional near-infrared spectroscopy (fNIRS), which measures changes in the concentrations of oxygenated hemoglobin ($\Delta HBO_2$) and deoxygenated haemoglobin ($\Delta HHB$)—similar to the blood oxygen level-dependent signal in fMRI. fNIRS is capable of non-invasive measurement of near-infrared light absorption within the range of 700 to 1000 nm through the skull [14]. In contrast to traditional MRI scanners, the portability and compatibility of fNIRS with ferromagnetic and electrical components provide researchers with the option to monitor and study functional brain activity in clinical settings [15,16].

Machine learning has played a pivotal role in neuroimaging-based methods for the study of pain [17,18]. It helps us to better understand the pain by uncovering patterns within clinical and experimental data [19]. Machine learning methods can effectively acquire the ability to map features to known classes, enabling them to predict a pain phenotype class based on a complex set of obtained features. For instance, Brown et al. [20], in an fMRI study, employed the Support Vector Machine (SVM) algorithm to distinguish between painful and non-painful experimental stimuli, achieving an accuracy of 81%. In an EEG study, Gram et al. [21] examined individuals who had received either morphine or a placebo following cold pressor test stimulation. They used the SVM algorithm to classify responders, achieving an accuracy of 71.9%. This classification was based on wavelet coefficients derived from each EEG band. These studies have shown the potential of neuroimaging and machine learning in the identification of pain.

In pain research using fNIRS, machine learning has proven to be effective for the detection and prediction of pain [22]. In a study by Pourshoghi et al. [22], authors used an SVM classifier using B-spline coefficients from functional data analysis. They achieved a classification accuracy of 94% in distinguishing between low-pain and high-pain signals using fNIRS. In Fernandez et al. [23], the results indicate that by using the Gaussian Support Vector Machine (SVM), they achieved an accuracy of 94.17% in classifying the four types of pain within the fNIRS data. Zeng et al. [24] investigated chronic pain's impact on brain function using fNIRS. Machine learning achieved high accuracy in identifying chronic pain patients based on resting-state fNIRS data, suggesting the potential for using functional connectivity features as neural markers for chronic pain diagnosis. Despite the promising

results obtained by the mentioned studies, there is still limited research in this field within the literature.

This study employs an approach for pain assessment that leverages the analysis of fNIRS signals in combination with machine learning techniques. This approach utilises fNIRS measurements of $\Delta HBO_2$ and $\Delta HHB$ to provide a comprehensive and accurate evaluation of pain levels. While the literature emphasises $\Delta HBO_2$ as a more promising fNIRS measure [25,26], recent studies, as highlighted by Ho et al. [27], indicate that both measures exhibit high accuracy in classification tasks. Therefore, in this study, both $\Delta HBO_2$ and $\Delta HHB$ measures have been taken into account. First, the pain information of 30 healthy subjects was collected using quantitative sensory testing (QST). Then, we performed a channel selection process to remove faulty channels from the analysis. Subsequently, ten statistical features from each measure were extracted. Then, we utilised well-known classifiers to identify pain levels using this reduced feature set. This study makes the following contributions: (1) proposing an fNIRS channel selection strategy for rejecting noisy channels based on high-frequency and high-amplitude artifacts; (2) presenting a group of possible features from fNIRS signals for the assessment of pain; (3) identifying that $\Delta HBO_2$ is better at detecting high pain intensity and $\Delta HHB$ is good at detecting low pain intensity; and (4) proposing the combination of $\Delta HBO_2$ and $\Delta HHB$ as a possible biomarker of human pain. This study contributes to the field of pain assessment and offers new avenues for understanding and quantifying pain in a more precise and objective manner.

## 2. Materials and Methods

Figure 1 presents the core system block diagram of the proposed fNIRS-based pain assessment system. The system integrates attributes from both $\Delta HBO_2$ and $\Delta HHB$ to assess the pain level. Further elaboration on the materials and methodology is provided in the following subsections.

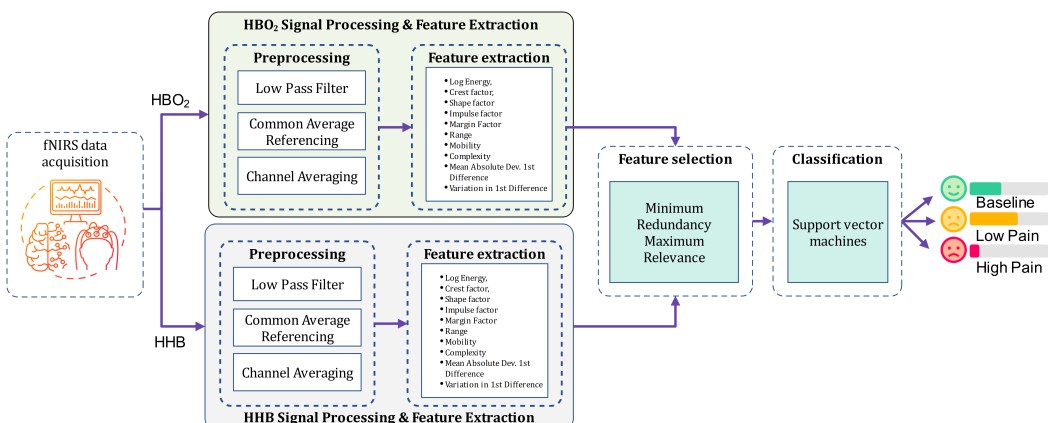

**Figure 1.** System block diagram of the proposed fNIRS-based pain assessment system.

### 2.1. Experimental Protocol

In this study, 30 healthy individuals (7 females and 23 males) aged 19 to 52 years ($31.7 \pm 8.7$ yrs) participated. None had unstable medical conditions, chronic pain, or recent medication usage prior to testing. Participants received detailed explanations and provided written informed consent before the start of the experiments. The research, involving human participants, received ethical approval from the University of Canberra's Human Research Ethics Committee (reference number 11837).

The data collection procedure took place at the Human–Machine Interface Laboratory at the University of Canberra, Australia. Participants were seated comfortably with both arms resting on the table. Electrodes from a transcutaneous electrical nerve stimulation (TENS) machine (Medihightec Medical Co., Ltd., Taipei City, Taiwan) were placed on the participants' inner forearm and the back of their right hand. The experimental process consisted of two phases: an initial assessment of individual pain perceptions using the

QST protocol, which determined pain thresholds and tolerances, followed by the pain stimulation phase. We defined the *pain threshold (low pain)* as the lowest stimulus intensity at which stimulation became painful, and *pain tolerance (high pain)* as the highest intensity of pain the participant could endure before reaching a point of intolerable discomfort. In the pain stimulation phase, fNIRS data were acquired and a 60 s baseline recording was obtained before the start of the experiment. A counterbalanced approach was employed, alternating between low and high stimuli intensity and forearm or hand stimulation. Six 10 s stimulus repetitions were recorded for each type of stimulus, followed by 40 s rest intervals. Figure 2 presents a schematic representation of the stimulation and perception of pain.

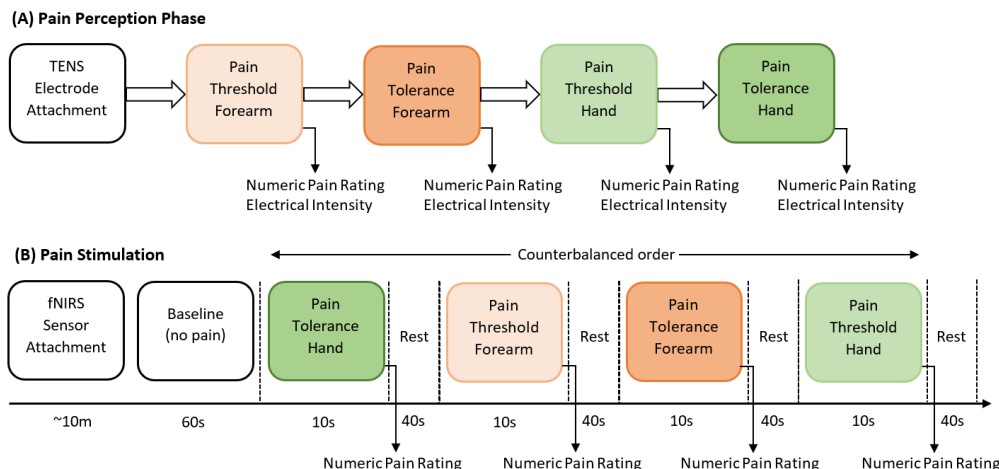

**Figure 2.** Schematic representation of the experimental procedure.

Changes in $\Delta HBO_2$ and $\Delta HHB$ concentration (µmol/L) were measured using a wireless, continuous wave fNIRS device (Artinis Medical Systems, Gelderland, the Netherlands). The fNIRS system includes 24 channels covering the prefrontal cortex (PFC). Optodes (10 sources and 8 detectors) are separated by 35 mm and placed on the frontal lobe (Figure 3). The near-infrared light was emitted by sources with wavelengths of 760 and 840 nm at a sampling rate of 50 Hz. Figure 4 displays the raw fNIRS channels ($\Delta HBO_2$) recorded over a 5 min duration while a subject experienced varying pain intensities.

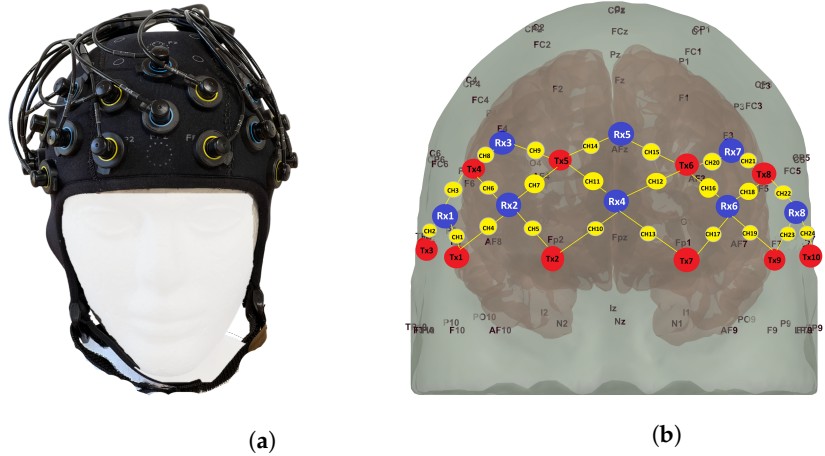

(**a**)  (**b**)

**Figure 3.** fNIRS channel information: (**a**) fNIRS cap. (**b**) Schematic of fNIRS channel locations. Red: Sources; Blue: Detectors; and Yellow: Channels. Specifically, the optodes Tx1, Tx2, Tx7, Tx9, Rx3, and Rx7 were positioned at the following locations on the standard 10–20 EEG system: Tx1: at F8; Tx2: at Fp2; Tx7: at Fp1; Tx9: at F7; Rx3: at F4; and Rx7: at F3.

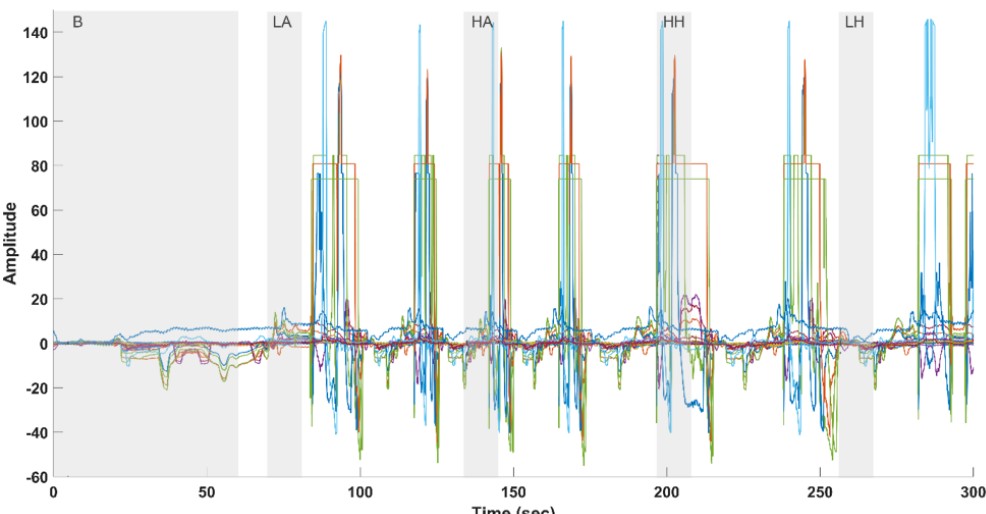

**Figure 4.** Twenty-two-Channel fNIRS (measuring changes in $\Delta HBO_2$) raw data (excluding two faulty channels) with annotated and highlighted durations for different conditions: B (Baseline), LA (Low Arm Pain), HA (High Arm Pain), HH (High Hand Pain), and LH (Low Hand Pain). The gray background in the figure represents the duration of each experiment phase: Baseline: 60 s, LA, LH, HA, and HH, each lasting 10 s.

### 2.2. Channel Selection

In the context of processing fNIRS data from 24 channels, as shown in Figure 4, some specific challenges arose. Two of the electrodes related to channels 19 and 23 were found to be malfunctioning, necessitating their exclusion from the analysis. This action was taken to ensure the integrity of the data. Additionally, among the remaining 22 channels, it was observed that certain channels exhibited distinct and undesirable features in the form of high amplitude and sharp peaks resembling square wave artifacts. These peculiar patterns suggest that these channels were significantly contaminated by movement artifacts or other non-neural artifacts. To effectively address this issue and proceed with data analysis, a preliminary step involved the systematic identification of unreliable channels to be excluded from further processing. This selection was accomplished using the relative range (RR) operator threshold. Relative range (Equation (1)) is defined as the ratio of the range of the derivative of an fNIRS channel to the range of the raw channel, as follows:

$$\text{RR} = \frac{\max(x'_{ch}) - \min(x'_{ch})}{\max(x_{ch}) - \min(x_{ch})} \tag{1}$$

where $x'_{ch}$ is the derivative of an fNIRS channel $x'_{ch}$, which represents the rate of change in a signal. In the context of fNIRS signals, the derivative can highlight regions where the signal changes rapidly, which may correspond to high-amplitude peaks (i.e., spikes) within a channel. As a result, high RR values indicate the presence of these high-amplitude sharp peaks. Experimental findings revealed that channels with an RR exceeding 0.1 (10%) are typically contaminated by these artifacts. With this threshold, the channels contaminated by artifacts were excluded, ensuring that only artifact-free channels were retained for subsequent processing. The raw fNIRS channels ($n = 17$) for $\Delta HBO_2$ measurement selected after the channel selection algorithm are shown in Figure 5. After this step, the data from three subjects were excluded from further processing as the algorithm resulted in the removal of over 70% of their number of channels. For the remaining 27 subjects, the number of retained channels after the selection process ranged from 16 to 22.

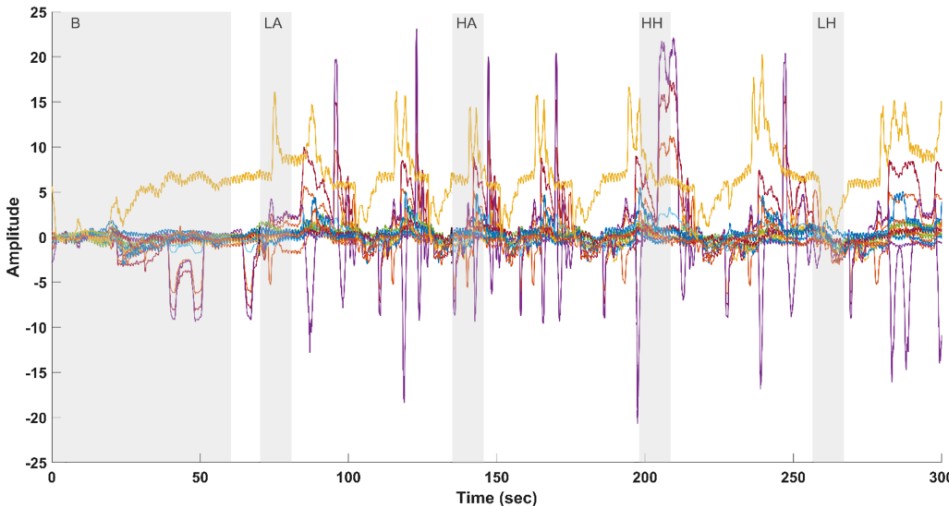

**Figure 5.** Raw fNIRS channels (measuring changes in $\Delta HBO_2$) selected after the proposed channel selection algorithm featuring the relative range (RR). The intervals for various pain conditions are highlighted and annotated as B (Baseline), LA (Low Arm Pain), HA (High Arm Pain), HH (High Hand Pain), and LH (Low Hand Pain). The gray background in the figure represents the duration of each experiment phase: Baseline: 60 s, LA, LH, HA, and HH, each lasting 10 s.

### 2.3. Dataset Organisation

After completing the data collection and channel selection process, all recorded data were segmented into 10-second intervals for each class. This resulted in six observations for the baseline class per subject, 12 observations for the low pain class per subject, and 12 observations for the high pain class per subject. In order to address the class observation imbalance, six additional observations from the rest periods of each subject, prior to the pain stimulation, were included in the baseline class. Consequently, the dataset consisted of a total of 972 observations. Each subject contributed 12 observations for each class, resulting in a cumulative total of 36 observations per subject. The dataset included 324 observations for each of the Baseline (B), Low Pain (LP), and High Pain (HP) classes.

### 2.4. Signal Processing: Filtration and Averaging

To suppress the noise and pulsation in fNIRS data ($\Delta HBO_2$ and $\Delta HHB$), as shown in Figure 6, each available fNIRS channel was passed through a 4th order Butterworth infinite impulse response low-pass filter with a cut-off frequency of 0.16 Hz [23].

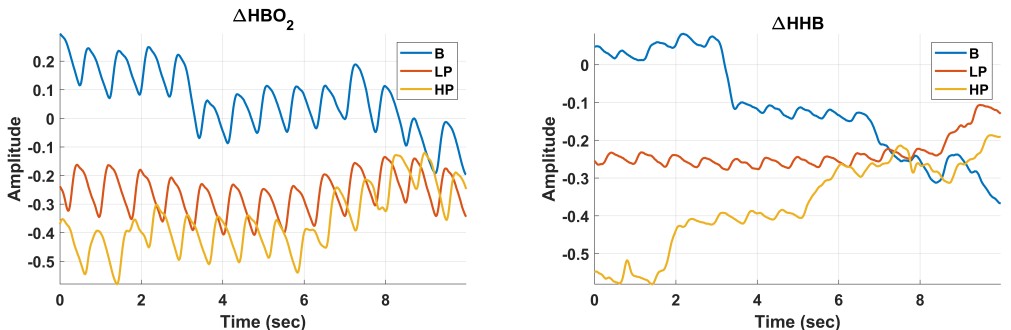

**Figure 6.** Raw 10-Second Data Segments for Baseline (B), Low Pain (LP), and High Pain (HP) Classes, displayed for Channel 1 of $\Delta HBO_2$ (**Left**) and $\Delta HHB$ (**Right**).

During fNIRS data acquisition, there can be various common noise sources that affect the measurements. These noise sources can include changes in blood flow unrelated to neural activity, motion artifacts, and systemic physiological changes such as heart rate and respiration [23]. These sources can introduce noise into the fNIRS data. Common Average

Referencing (CAR) [28] involves calculating the averages from all available channels across the scalp for each wavelength ($\Delta HBO_2$ and $\Delta HHB$). This average is then subtracted, for each wavelength from the signal of each individual channel. This effectively subtracts out the common noise components shared by all channels. Equation (2) shows the channel-averaging scheme:

$$h_{\text{avg}}(k) = \frac{1}{M} \sum_{j=1}^{M} H(k, j) \tag{2}$$

where $h$ is the average of fNIRS measure $H$ ($\Delta HBO_2$ or $\Delta HHB$), $M$ is the total number of channels for each participant, $k$ is the discrete time for which the signal is recorded, and $j$ is the channel number. The preprocessed version of both fNIRS measures, i.e., $\Delta HBO_2$ and $\Delta HHB$ for various experimental conditions, is displayed in Figure 7.

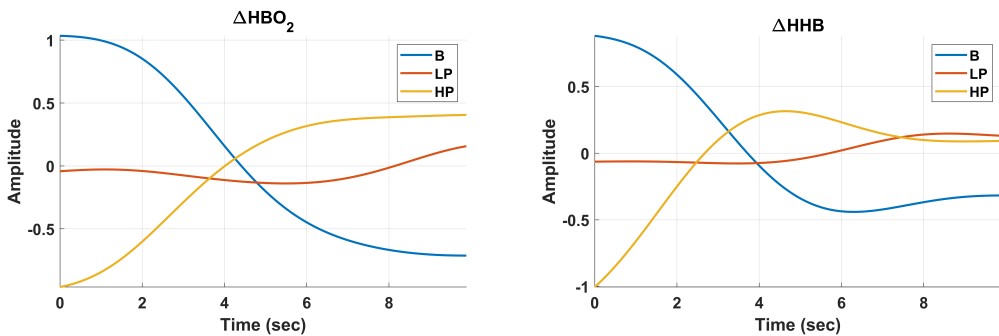

**Figure 7.** Preprocessed 10-Second Data Segments for Baseline (B), Low Pain (LP), and High Pain (HP) Classes, displayed for $\Delta HBO_2$ (**Left**) and $\Delta HHB$ (**Right**). The processing pipeline encompasses low-pass filtering, Common Average Referencing (CAR) for each filtered channel, and the final step of averaging across all channels, culminating in a consolidated vector representation.

### 2.5. Feature Extraction

The $\Delta HBO_2$ and $\Delta HHB$ signals display distinct characteristics associated with the pain intensities. Amplitude, as an indicator of pain intensity, increases with more painful stimuli, signifying higher neural activity and oxygen demand. Variation in these signals highlights the dynamic nature of pain experiences, showcasing rapid and substantial fluctuations over time. Complexity in $\Delta HBO_2$ and $\Delta HHB$ responses uncovers the intricate interactions between brain regions and physiological systems involved in pain processing [29]. The dynamics of $\Delta HBO_2$ and $\Delta HHB$ responses reveal the timing of pain intensity, from pain onset to apex, and then, to recovery. Moreover, the stability of these signals distinguishes sustained pain from transient changes, providing insights into the persistence of pain perception. To extract the fNIRS signal information related to intensity, dynamics, stability, complexity, and variation-like characteristics [30], we have carefully chosen features [31,32] such as Log Energy, Crest Factor, Shape Factor, Impulse Factor, Margin Factor, Mobility, Complexity, Mean Absolute Deviation of First Difference, Range, and Variation in First Difference as defined in Table 1. These features are extracted from both $\Delta HBO_2$ and $\Delta HHB$ signals and fused at the feature level to create a fused feature vector.

### 2.6. Feature Selection

Feature selection is crucial for improving model efficiency by focusing on important features, reducing dimensionality, and ultimately improving the overall performance in machine learning tasks. In this work, the *Minimum Redundancy Maximum Relevance (MRMR)* algorithm [33] is utilised. MRMR identifies the most informative features for a given task by considering both their relevance to the target variable and their redundancy with respect to each other. It evaluates the mutual information between features and the target, ranking them by relevance while also measuring the redundancy between features. The algorithm

then selects features that achieve the right balance between relevance and redundancy, resulting in a subset of features that can improve model performance with reduced features.

**Table 1.** Details of statistical features used in this study. The feature vector $F$ comprises all ten features, with $h$ as the preprocessed signal ($\Delta HBO_2$ or $\Delta HHB$), $h'$ as the derivative of $h$, and $\bar{h}'$ as the mean of $h'$. $h_{peak}$, $h_{rms}$, and $h_{am}$ denote the peak, root mean square, and absolute mean of the input signal $h$, respectively, while $var(.)$ represents the variance.

| Features | Definitions |
|---|---|
| Log Energy | $F_1 = \sum_{i=1}^{n} log(h_i^2)$ |
| Crest Factor | $F_2 = \frac{h_{peak}}{h_{rms}}$ |
| Shape Factor | $F_3 = \frac{h_{rms}}{h_{am}}$ |
| Impulse Factor | $F_4 = \frac{h_{peak}}{h_{am}}$ |
| Margin Factor | $F_5 = \frac{h_{peak}}{h_{am}^2}$ |
| Mobility | $F_6 = \sqrt{\frac{var(h')}{var(h)}}$ |
| Complexity | $F_7 = \frac{F_6(h')}{F_6(h)}$ |
| Mean Absolute Deviation of First Derivative | $F_8 = \frac{1}{n} \sum_{i=1}^{n} |h_i' - \bar{h}'|$ |
| Range | $F_9 = \max(h) - \min(h)$ |
| Variation in First Derivative | $F_{10} = \sqrt{\frac{1}{n} \sum_{i=1}^{n} (h_i' - \bar{h}')^2}$ |

### 2.7. Classification

In the context of pain level assessment, the classification focus was on distinguishing between various pain classes: Baseline (B), Low Pain (LP), and High Pain (HP). To achieve this, we employed a reduced feature set consisting of statistical features extracted from both $\Delta HBO_2$ and $\Delta HHB$ signals. We utilised well-known classifiers such as Discriminant (Disc) [34], K-Nearest Neighbour (KNN) [35], and Support Vector Machine (SVM) [36] to identify pain levels using the feature set. We employed parameter optimisation, carefully tuning the classifiers using a Bayesian approach [37]. This data-driven decision-making process is supported by an acquisition function known as 'expected improvement per second plus', which underwent 50 iterations. We identified the hyperparameters for each classification algorithm that minimised the 10-fold cross-validation loss across the entire dataset [38].

The classification performance was evaluated using a leave-one-subject-out cross-validation (LOSOCV) approach [39]. In LOSOCV, the model's effectiveness is assessed by withholding one individual's data from the dataset for testing, while the data from the remaining participants undergoes 10-fold cross validation. This process is repeated iteratively for each subject in the dataset, ensuring that each subject serves as the test set exactly once. The performance metrics consisting of accuracy (Acc), sensitivity (Sen), specificity (Spec), and F1 score (F1) and obtained in each iteration were averaged to provide a comprehensive assessment of the model's overall performance. Additionally, we systematically tested the identification of the best-performing model with varying numbers of features based on their MRMR rank. Thus, the combination of feature engineering, hyperparameter optimisation, and classification algorithms proves to be a powerful toolkit for decoding pain levels based on fNIRS signals.

### 2.8. Statistical Analysis

The obtained features were also analysed using statistical analysis to identify significant differences in the obtained features across the different experimental conditions for both $\Delta HBO_2$ and $\Delta HHB$ independently. This information will help validate our hypothesis, indicating that the obtained features encompass pain-related data from the experimental conditions. First, the data were examined for normality and homogeneity using the Kolmogorov–Smirnov tests. Focusing on the ten extracted features from $\Delta HBO_2$ and

$\Delta HHB$ measurements for the classification of the pain level, differences were analysed using Analysis of Variance (ANOVA). A post hoc Bonferroni test was carried out for multiple comparisons. The significant level was set to $p < 0.05$. All statistical analyses were performed using SPSS version 29.

## 3. Results

In this section, the outcomes of the proposed multi-class fNIRS-based pain assessment system are presented. The results of the system are demonstrated via the independent utilisation of $\Delta HBO_2$ and $\Delta HHB$ signals, along with employing combined haemoglobin measures. Ten features are extracted from each measure and are passed to the three classifiers (Disc, KNN, and SVM). In the case of $\Delta HBO_2 + \Delta HHB$, the features from each measure are fused before the classification stage, resulting in a total of 20 features in this case. The selection of classifiers for each experiment was made following extensive hyperparameter tuning, as detailed in Table 2.

Activation levels of fNIRS using both $\Delta HBO_2$ and $\Delta HHB$ measurements for different experimental conditions are presented in Figure 8. As shown, the highest activation in the prefrontal cortex for $\Delta HBO_2$ (first row) is recorded for HA (High Arm pain), while LH (Low Arm pain) exhibits the lowest concentration level compared to other conditions and with a very similar activation level to the baseline. Similar to $\Delta HBO_2$, the most elevated activation in $\Delta HHB$ measures is observed in the HA condition. However, other conditions do not exhibit a significant increase.

**Table 2.** Optimised hyperparameters for different classification algorithms via Bayesian Optimisation in the context of distinguishing between Baseline (B), Low Pain (LP), and High Pain (HP).

| Model | Parameters | $\Delta HBO_2$ | $\Delta HHB$ | $\Delta HBO_2 + \Delta HHB$ |
|---|---|---|---|---|
| Disc | Discriminant Type | Pseudo Linear | Linear | Diagonal Linear |
| | Gamma | $7.55 \times 10^{-4}$ | 0.0025 | 0.006 |
| | Delta | $3.51 \times 10^{-5}$ | $2.96 \times 10^{-5}$ | $2.12 \times 10^{-5}$ |
| KNN | Number of Neighbours | 211 | 1 | 25 |
| | Distance | Chebychev | Cosine | City Block |
| | Distance Weight | Inverse | Inverse | Equal |
| | Exponent | – | – | – |
| | Neighbour Search | KD-Tree | Exhaustive | Exhaustive |
| | Standardisation | Yes | Yes | Yes |
| SVM | Coding | One vs. All | One vs. All | One vs. One |
| | Box Constraint | 2.1888 | 10.3923 | 980.4894 |
| | Kernel Scale | – | – | 13.2018 |
| | Kernel Function | Polynomial | Polynomial | Gaussian |
| | Polynomial Order | 3 | 3 | – |
| | Standardise | Yes | Yes | Yes |

### 3.1. Classification Results

The results in terms of performance metrics for each measure are presented in Table 3. For the $\Delta HBO_2$ measure, the SVM classifier performs remarkably well as compared to that of Disc and KNN, achieving the highest accuracy of 64.67%. It exhibits outstanding sensitivity (92.85%) and specificity (97.22%), underlining its ability to effectively identify pain instances while maintaining high precision. For the $\Delta HHB$ measure, the SVM classifier again excels with the highest accuracy of 62.28%. It maintains remarkable sensitivity (92.87%) and specificity (97.07%), showcasing its effectiveness in pain assessment. The KNN classifier exhibits an accuracy of 41.83%, whereas the Disc classifier displays an accuracy of 50.94%. The combined $\Delta HBO_2 + \Delta HHB$ measure, when paired with the SVM classifier, outperforms other classification algorithms with an accuracy of 66.55%. Sensitivity (93.8%) and specificity (96.14%) remain high, highlighting the SVM's effectiveness in pain assessment. The F1 Score of 96.98% emphasises the balanced performance. On the other

hand, the KNN classifier, with an accuracy of 40.19%, shows lower performance, and the Disc classifier, with an accuracy of 56.23%, exhibits moderate performance. The SVM classifier consistently achieves high accuracy, sensitivity, specificity, and F1 Score, with the $\Delta HBO_2 + \Delta HHB$ measure performing the best among all measures.

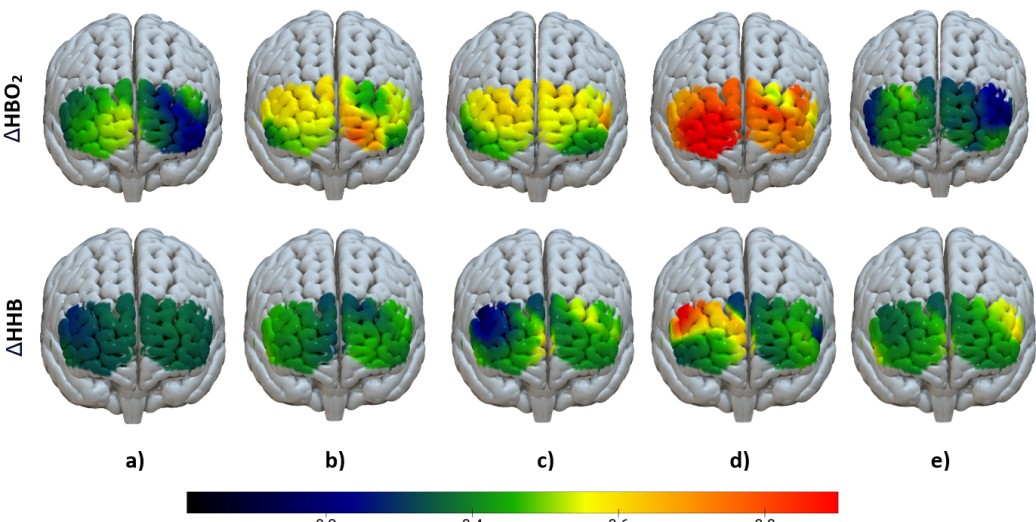

**Figure 8.** Haemodynamic changes shown using fNIRS for $\Delta HBO_2$ (first row) and $\Delta HHB$ (second row) measures: (**a**) Baseline, (**b**) HH (High Hand Pain), (**c**) LH (Low Hand Pain), (**d**) HA (High Arm Pain), and (**e**) LA (Low Arm Pain). The color bar signifies the change in concentration of $\Delta HBO_2$ and $\Delta HHB$ (Δμmol). These calculations are derived from the averages across all subjects for each respective channel.

**Table 3.** System performance metrics (Acc: Accuracy, Sen: Sensitivity, Spec: Specificity, and F1 Score) for different classification algorithms (Disc, KNN, and SVM) across various measures, with each measure having a different feature vector length denoted by #.

| Measure | Model | # | Acc | Sen | Spec | F1 Score |
|---|---|---|---|---|---|---|
| | Disc | | $51.78 \pm 9.94$ | $74.78 \pm 19.43$ | $73.30 \pm 11.91$ | $85.98 \pm 9.60$ |
| $\Delta HBO_2$ | KNN | 10 | $41.74 \pm 7.73$ | $56.96 \pm 17.03$ | $64.81 \pm 13.29$ | $75.53 \pm 7.85$ |
| | SVM | | $64.67 \pm 5.99$ | $92.85 \pm 8.25$ | $97.22 \pm 3.65$ | $96.67 \pm 3.76$ |
| | Disc | | $50.94 \pm 7.60$ | $73.57 \pm 12.12$ | $75.77 \pm 9.53$ | $85.35 \pm 6.42$ |
| $\Delta HHB$ | KNN | 10 | $41.83 \pm 8.34$ | $44.36 \pm 14.30$ | $74.23 \pm 9.10$ | $73.14 \pm 5.75$ |
| | SVM | | $62.28 \pm 5.86$ | $92.87 \pm 8.24$ | $97.07 \pm 3.62$ | $96.63 \pm 3.83$ |
| | Disc | | $56.23 \pm 6.84$ | $76.32 \pm 11.62$ | $79.32 \pm 10.81$ | $87.24 \pm 5.73$ |
| $\Delta HBO_2 + \Delta HHB$ | KNN | 20 | $40.19 \pm 8.09$ | $43.63 \pm 13.77$ | $68.06 \pm 10.01$ | $70.99 \pm 6.07$ |
| | SVM | | $66.55 \pm 7.36$ | $93.8 \pm 6.38$ | $96.14 \pm 3.04$ | $96.98 \pm 3.08$ |

Following the acquisition of reference values using the full feature set (see Table 3), the feature set underwent a feature selection process using MRMR to minimise redundancy and enhance the discriminative power. Table 4 presents the performance metrics for each measure after applying MRMR. The results provide insights into how feature selection impacts the performance of pain assessment models. In the $\Delta HBO_2$ measure, the feature selection process has notably influenced the performance of different classifiers. The SVM classifier, with nine selected features, achieves the highest accuracy of 65.71% with improved sensitivity (93.18%) and specificity (95.99%). The KNN classifier, with seven selected features, exhibits enhanced accuracy at 44.22%, although it still falls behind SVM.

In the $\Delta HHB$ measure, feature selection has similarly enhanced the performance of the classifiers. The SVM classifier, with nine selected features, maintains its position as the top-performing classifier with an accuracy of 63.42% along with improved sensitivity (94.44%) and specificity (97.22%). The combined $\Delta HBO_2 + \Delta HHB$ measure benefits from

feature selection, particularly in the SVM classifier with 15 selected features. It achieves the highest accuracy at 68.51%, emphasising the significance of choosing both hemoglobin measures. Sensitivity (94.7%), specificity (94.29%), and the F1 Score also reflect notable improvements. The KNN classifier, with 18 selected features, shows an accuracy of 40.8%. These findings emphasise the crucial role of both measures, particularly in the combined ($\Delta HBO_2 + \Delta HHB$) measure, where the SVM classifier emerges as the optimal choice for precise and well-balanced pain assessment.

**Table 4.** System performance metrics (Acc: Accuracy, Sen: Sensitivity, Spec: Specificity, and F1 Score) with MRMR-based selected features for different classification algorithms (Disc, KNN, and SVM) applied to each measure, with the feature vector length denoted by #.

| Measure | Model | # | Acc | Sen | Spec | F1 Score |
|---|---|---|---|---|---|---|
| $\Delta HBO_2$ | Disc | 10 | 51.78 ± 9.94 | 74.78 ± 19.43 | 73.30 ± 11.91 | 85.98 ± 9.60 |
| | KNN | 7 | 44.22 ± 8.16 | 55.36 ± 15.38 | 70.22 ± 13.16 | 76.30 ± 7.20 |
| | SVM | 9 | 65.71 ± 5.97 | 93.18 ± 8.03 | 95.99 ± 4.24 | 96.77 ± 3.67 |
| $\Delta HHB$ | Disc | 10 | 50.94 ± 7.6 | 73.57 ± 12.12 | 75.77 ± 9.53 | 85.35 ± 6.42 |
| | KNN | 10 | 41.83 ± 8.34 | 44.36 ± 14.3 | 74.23 ± 9.10 | 73.14 ± 5.75 |
| | SVM | 9 | 63.42 ± 6.85 | 94.44 ± 8.33 | 97.22 ± 3.27 | 97.40 ± 3.84 |
| $\Delta HBO_2 + \Delta HHB$ | Disc | 20 | 56.23 ± 6.84 | 76.32 ± 11.62 | 79.32 ± 10.81 | 87.24 ± 5.73 |
| | KNN | 18 | 40.8 ± 7.26 | 44.58 ± 15.27 | 68.83 ± 9.34 | 71.72 ± 6.34 |
| | SVM | 15 | 68.51 ± 9.02 | 94.70 ± 5.77 | 94.29 ± 4.92 | 97.33 ± 2.92 |

Table 5 lists the features corresponding to the optimal results for each measure. In the approach using a fusion of haemoglobin measures ($\Delta HBO_2 + \Delta HHB$), among the fifteen selected features, nine belong to $\Delta HBO_2$, highlighting its greater contribution compared to the six features from $\Delta HHB$.

**Table 5.** List of selected features for each measure, with # indicating the number of features.

| Measure | # | Selected Features |
|---|---|---|
| $\Delta HBO_2$ | 9 | Mobility, Complexity, Range, Shape Factor, Variation in First Derivative, Impulse Factor, Mean Absolute Deviation of First Derivative, Log Energy, Crest Factor. |
| $\Delta HHB$ | 9 | Crest Factor, Complexity, Shape Factor, Mobility, Range, Variation in First Derivative, Log Energy, Mean Absolute Deviation of First Derivative, Margin Factor. |
| $\Delta HBO_2 + \Delta HHB$ | 15 | $\Delta HBO_2$: Mobility, Complexity, Range, Shape Factor, Variation in First Derivative, Impulse Factor, Mean Absolute Deviation of First Derivative, Log Energy, Crest Factor.<br>$\Delta HHB$: Crest Factor, Complexity, Shape Factor, Mobility, Range, Variation in First Derivative. |

In pain assessment, class-wise performance is also crucial because it enables the accurate identification of different pain levels, helping clinicians in making treatments based on individual pain experiences and needs. Analysing the class-wise performance of each measure, as depicted in Figure 9, highlights the superior effectiveness of the SVM classifier, particularly in accurately classifying instances of Baseline (B), Low Pain (LP), and High Pain (HP) compared to other classification methods such as Disc and KNN. Notably, the $\Delta HBO_2$ measure demonstrates its strength in achieving higher classification accuracy for High Pain (HP) instances, while the $\Delta HHB$ measure excels in classifying Low Pain (LP) cases. However, it is important to emphasise the significance of identifying the absence of pain (B) in pain assessment, and here, the $\Delta HHB$ measure proves better at predicting pain-free observations compared to the $\Delta HBO_2$ measure. The fusion of both $\Delta HBO_2$ and $\Delta HHB$ effectively integrates this information, yielding improved results for

both LP and HP classes. In summary, the fusion of both fNIRS measures enhances class-wise accuracies in pain assessment, contributing to a more comprehensive and precise pain perception evaluation.

**$\Delta HBO_2$ — Disc**

| | B | LP | HP |
|---|---|---|---|
| B | 73.77 | 13.89 | 11.11 |
| LP | 26.85 | 35.8 | 37.35 |
| HP | 26.54 | 28.4 | 45.06 |

**$\Delta HHB$ — Disc**

| | B | LP | HP |
|---|---|---|---|
| B | 72.53 | 11.73 | 14.51 |
| LP | 22.22 | 41.05 | 36.73 |
| HP | 26.23 | 35.19 | 38.58 |

**$\Delta HBO_2 + \Delta HHB$ — Disc**

| | B | LP | HP |
|---|---|---|---|
| B | 75.31 | 12.35 | 11.11 |
| LP | 18.52 | 45.68 | 35.8 |
| HP | 22.84 | 30.25 | 46.91 |

**$\Delta HBO_2$ — KNN**

| | B | LP | HP |
|---|---|---|---|
| B | 54.63 | 20.99 | 23.15 |
| LP | 29.63 | 30.56 | 39.81 |
| HP | 29.94 | 23.15 | 46.91 |

**$\Delta HHB$ — KNN**

| | B | LP | HP |
|---|---|---|---|
| B | 43.83 | 27.78 | 27.16 |
| LP | 27.78 | 38.89 | 33.33 |
| HP | 23.77 | 33.95 | 42.28 |

**$\Delta HBO_2 + \Delta HHB$ — KNN**

| | B | LP | HP |
|---|---|---|---|
| B | 44.14 | 30.56 | 24.07 |
| LP | 34.57 | 41.67 | 23.77 |
| HP | 27.78 | 36.11 | 36.11 |

**$\Delta HBO_2$ — SVM**

| | B | LP | HP |
|---|---|---|---|
| B | 91.98 | 1.235 | 5.556 |
| LP | 4.012 | 45.06 | 50.93 |
| HP | 4.012 | 36.73 | 59.26 |

**$\Delta HHB$ — SVM**

| | B | LP | HP |
|---|---|---|---|
| B | 93.21 | 2.469 | 3.086 |
| LP | 3.086 | 52.16 | 44.75 |
| HP | 2.469 | 53.4 | 44.14 |

**$\Delta HBO_2 + \Delta HHB$ — SVM**

| | B | LP | HP |
|---|---|---|---|
| B | 93.52 | 2.16 | 3.086 |
| LP | 5.864 | 54.63 | 39.51 |
| HP | 5.556 | 37.96 | 56.48 |

**Figure 9.** Class-wise accuracy (%) assessment of different measures using Disc; KNN; and SVM classifiers using confusion charts.

### 3.2. Statistical Analyses

The results regarding the comparison of the statistically significant $\Delta HBO_2$ feature in the different experimental conditions are provided in Table 6. Among ten different features for $\Delta HBO_2$ measurement, Log Energy, Crest Factor, Shape Factor, and Range exhibit significant differences compared to other features in distinguishing between experiment conditions, as indicated by their respective *p*-values ($F_{(2,972)} = 3.078$, $p = 0.046$, $F_{(2,972)} = 3.264$, $p = 0.039$, $F_{(2,972)} = 3.466$, $p = 0.032$, $F_{(2,972)} = 10.179$, $p < 0.001$, respectively). For $\Delta HHB$ measures, three features, Log Energy, Margin Factor, and Range, showed significant differences in identifying pain levels as compared to other features ($F_{(2,972)} = 3.127$, $p = 0.044$, $F_{(2,972)} = 4.134$, $p = 0.016$, $F_{(2,972)} = 4.558$, $p = 0.011$, respectively). The results of the post hoc test for the comparison of pain levels for statistically significant features of the $\Delta HHB$ measure have been provided in Table 7.

**Table 6.** Post Hoc Test Results for Different Levels of Pain in Various Features of $\Delta HBO_2$ (Only comparisons with significant ($p \leq 0.05$) values are reported.)

| Feature | Group One | Group Two | Mean Diff. | Std. Error | Sig. | Lower Bound | Upper Bound |
|---|---|---|---|---|---|---|---|
| Log Energy | No Pain | Low Pain | 73.67 | 41.110 | 0.073 | −7.00 | 154.33 |
| | | High Pain | 96.99 * | 41.110 | 0.018 | 16.33 | 177.66 |
| | Low Pain | No Pain | −73.67 | 41.110 | 0.073 | −154.33 | 7.00 |
| | | High Pain | 23.33 | 40.966 | 0.569 | −57.06 | 103.71 |
| | High Pain | No Pain | −96.99 * | 41.110 | 0.018 | −177.66 | −16.33 |
| | | Low Pain | −23.33 | 40.966 | 0.569 | −103.71 | 57.06 |
| Crest factor | No Pain | Low Pain | −0.02 | 0.039 | 0.685 | −0.09 | 0.06 |
| | | High Pain | 0.078 * | 0.039 | 0.044 | 0.00 | 0.15 |
| | Low Pain | No Pain | 0.02 | 0.039 | 0.685 | −0.06 | 0.09 |
| | | High Pain | 0.094 * | 0.039 | 0.015 | 0.02 | 0.17 |
| | High Pain | No Pain | −0.078 * | 0.039 | 0.044 | −0.15 | 0.00 |
| | | Low Pain | −0.094 * | 0.039 | 0.015 | −0.17 | -0.02 |
| Shape factor | No Pain | Low Pain | −0.017 * | 0.006 | 0.008 | −0.03 | 0.00 |
| | | High Pain | −0.01 | 0.006 | 0.067 | −0.02 | 0.00 |
| | Low Pain | No Pain | 0.017 * | 0.006 | 0.008 | 0.00 | 0.03 |
| | | High Pain | 0.01 | 0.006 | 0.402 | −0.01 | 0.02 |
| | High Pain | No Pain | 0.01 | 0.006 | 0.067 | 0.00 | 0.02 |
| | | Low Pain | −0.01 | 0.006 | 0.402 | −0.02 | 0.01 |

**Table 6.** *Cont.*

| Feature | Group One | Group Two | Mean Diff. | Std. Error | Sig. | Lower Bound | Upper Bound |
|---|---|---|---|---|---|---|---|
| | No Pain | Low Pain | −0.05 | 0.055 | 0.386 | −0.16 | 0.06 |
| | | High Pain | 0.08 | 0.055 | 0.167 | −0.03 | 0.19 |
| Impulse factor | Low Pain | No Pain | 0.05 | 0.055 | 0.386 | −0.06 | 0.16 |
| | | High Pain | 0.125 * | 0.055 | 0.024 | 0.02 | 0.23 |
| | High Pain | No Pain | −0.08 | 0.055 | 0.167 | −0.19 | 0.03 |
| | | Low Pain | −0.125 * | 0.055 | 0.024 | −0.23 | −0.02 |
| | No Pain | Low Pain | −0.165 * | 0.037 | $p \le 0.001$ | −0.24 | −0.09 |
| | | High Pain | −0.129 * | 0.037 | 0.001 | −0.20 | −0.06 |
| Range | Low Pain | No Pain | 0.165 * | 0.037 | $p \le 0.001$ | 0.09 | 0.24 |
| | | High Pain | 0.04 | 0.037 | 0.33 | −0.04 | 0.11 |
| | High Pain | No Pain | 0.129 * | 0.037 | 0.001 | 0.06 | 0.20 |
| | | Low Pain | −0.04 | 0.037 | 0.33 | −0.11 | 0.04 |

*: the mean difference is significant at 0.05 level.

**Table 7.** Post Hoc Test Results for Different Levels of Pain in Various Features of $\Delta HHB$ (Only comparisons with significant *p*-values are reported).

| Feature | Group One | Group Two | Mean diff. | Std. Error | Sig. | Lower Bound | Upper Bound |
|---|---|---|---|---|---|---|---|
| | No Pain | Low Pain | 104.153 * | 49.629 | 0.036 | 6.770 | 201.535 |
| | | High Pain | 110.774 * | 49.629 | 0.026 | 13.391 | 208.156 |
| Log Energy | Low Pain | No Pain | −104.153 * | 49.629 | 0.036 | −201.535 | −6.770 |
| | | High Pain | 6.62 | 49.4554 | 0.894 | −90.420 | 103.662 |
| | High Pain | No Pain | −110.774 * | 49.629 | 0.026 | −208.156 | −13.391 |
| | | Low Pain | −6.62 | 49.455 | 0.894 | −103.662 | 90.420 |
| | No Pain | Low Pain | 1.629 * | 0.572 | 0.004 | 0.506 | 2.752 |
| | | High Pain | 0.621 | 0.572 | 0.277 | −0.501 | 1.744 |
| Margin Factor | Low Pain | No Pain | −1.629 * | 0.572 | 0.004 | −2.752 | −0.506 |
| | | High Pain | −1.007 | 0.570 | 0.078 | −2.126 | 0.111 |
| | High Pain | No Pain | −0.621 | 0.572 | 0.277 | −1.744 | 0.501 |
| | | Low Pain | 1.007 | 0.570 | 0.078 | −0.111 | 2.126 |
| | No Pain | Low Pain | −0.106 * | 0.04 | 0.00 | −0.18 | −0.04 |
| | | High Pain | −0.072 * | 0.04 | 0.04 | −0.14 | 0.00 |
| Range | Low Pain | No Pain | 0.106 * | 0.04 | 0.00 | 0.04 | 0.18 |
| | | High Pain | 0.03 | 0.04 | 0.34 | −0.04 | 0.10 |
| | High Pain | No Pain | 0.072 * | 0.04 | 0.04 | 0.00 | 0.14 |
| | | Low Pain | −0.03 | 0.04 | 0.34 | −0.10 | 0.04 |

*: the mean difference is significant at 0.05 level.

## 4. Discussions

To the best of the authors' knowledge, this is the first study that deals with the objective assessment of pain via fNIRS within a comprehensive exploration of $\Delta HBO_2$ and $\Delta HHB$ measures. The findings reveal an association between pain intensities and distinct statistical patterns in haemoglobin concentrations. Considering the overall system accuracy, the $\Delta HBO_2$ measure demonstrated better performance than the $\Delta HHB$ measure in the multiclass scenario used in this study. However, when examining accuracies for specific classes, $\Delta HBO_2$ excels in identifying High Pain signals, while $\Delta HHB$ demonstrates better accuracy for Low Pain observations. Upon a comparison of both fNIRS measures, it can be concluded that the fusion of $\Delta HBO_2$ and $\Delta HHB$ measures at the feature level emerges as an effective method for the categorisation of the three pain intensities in our experimental conditions.

Based on the classification results, it can be deduced that the SVM classification algorithm is most effective when used with the selected statistical features across all the measures in pain assessment. Both of the fNIRS measures are considered to be reliable in evaluating pain, with $\Delta HBO_2$ demonstrating slightly higher accuracy than the $\Delta HHB$ measure when used independently. However, the most optimal results are obtained when combining both $\Delta HBO_2$ and $\Delta HHB$, suggesting that a combination of these two measures offers the best performance for pain assessment in our experimental conditions. While $\Delta HBO_2$ provides insights into the oxygenated haemoglobin concentration, which can indicate changes in blood flow and tissue activity, $\Delta HHB$ reveals deoxygenated haemoglobin

levels reflecting variations in tissue oxygen consumption. By integrating these two measures, a more holistic understanding of the physiological responses to pain is achieved. This combined approach allows for a more robust assessment as it captures both the supply and demand aspects of oxygen delivery, thus enhancing the ability to detect and interpret changes in pain perception.

Existing studies on pain assessment using neuroimaging methods have primarily focused on binary classifications, mainly distinguishing between pain and no pain. However, the development of approaches capable of distinguishing various signatures of pain has been neglected so far. This limitation is significant given the diverse origins (e.g., peripheral, emotional, and phantom pain), varying intensities, and durations of pain experienced in the human body. Different types of pain are carried to the central nervous system by different sensory receptors, responding to various stimuli associated with pain, such as temperature, chemical, or pressure [15]. Hence, there is a need for machine learning models that can effectively differentiate between multiple pain signatures at varying intensities, offering greater relevance for real-world scenarios. In contrast, our study addresses this gap by focusing on multilevel pain classification, considering pain originating from different locations of the body, specifically the hand and arm. This is particularly important for patients who are unable to communicate verbally, such as elderly people recovering from a stroke or with advanced dementia, and when the source of pain is not readily apparent.

In examining activation levels across different pain conditions, our focus on $\Delta HBO_2$ and $\Delta HHB$ measures provides valuable insights into the neural responses associated with pain perception. As depicted in Figure 8, the most pronounced increase in $\Delta HBO_2$ levels occurs in response to High Arm (HA) pain, emphasising the sensitivity of this measure to high pain intensities. Similarly, increased activation is evident in High Hand (HH) pain compared to Low Hand (LH) and Low Arm (LA) pains. These outcomes are consistent with prior research, reinforcing the notion that elevated pain levels are associated with a more pronounced neural response in fNIRS studies investigating $\Delta HBO_2$ [22,40]. The significant increase in activation during various conditions compared to the baseline, except for Low Arm (LA) pain, is noteworthy. This could, in part, account for the lower accuracy observed in identifying the low pain (LP) class using $\Delta HBO_2$ across all three classifiers. Activation levels in $\Delta HHB$ data exhibited minimal fluctuations across diverse conditions, with the exception of the HA condition, where the highest activation, akin to $\Delta HBO_2$, was observed. These findings highlight the superior efficacy of $\Delta HBO_2$ as a more reliable measure for pain assessment compared to $\Delta HHB$, when used independently. However, when combined in a feature fusion scheme, they collectively obtained better accuracy than when used independently.

While our proposed system demonstrated better performance in identifying different pain levels, it presents some limitations. First, the channel selection algorithm employed in our study served the purpose of rejecting channels saturated with artifacts and noise. However, it may automatically discard channels containing valuable pain-related information. To address this, a more advanced preprocessing algorithm should be considered, capable of mitigating noise in unreliable channels without outright rejection. This would ensure that potentially relevant information is retained in the dataset for more comprehensive pain assessment. Second, it is evident in our preprocessing stage, where we opted to average out all channels to generate a single time series vector. This approach, while simplifying the data, has the drawback of suppressing information inherent in individual channels. In our future work, we will conduct analysis by defining specific regions of interest based on functional areas of the brain, which can provide insights into the localised functions and responses to pain associated with different brain regions. Finally, our investigation into fNIRS data primarily focused on the time domain, emphasising the extraction and assessment of simple statistical features. However, by exclusively focusing on the time domain, we may have overlooked valuable information present in other domains. To broaden the scope of the analysis, we should consider additional domains, such as frequency or cepstral domains, throughout the stages of preprocessing, feature extraction, and evaluation.

## 5. Conclusions

In this study, we introduced a multilevel pain intensity assessment using fNIRS data, compiling a novel dataset from healthy individuals experiencing varying induced pain levels in distinct body locations. Analysing $\Delta HBO_2$ and $\Delta HHB$ measures, we found that $\Delta HBO_2$ outperformed $\Delta HHB$ overall but excelled in predicting high and low pain classes, respectively. Combining both measures significantly improved the performance, demonstrating the potential of fNIRS for multilevel pain assessment. The system achieved $68.51\% \pm 9.02\%$ accuracy, $94.7\% \pm 5.77\%$ sensitivity, and $94.29\% \pm 4.92\%$ specificity in diagnosing no pain, low pain, and high pain observations, respectively. Future research aims to explore integrating fNIRS with other sensor modalities, analysing pain-related information in different fNIRS domains, and effectively pinpointing the site of pain.

**Author Contributions:** Conceptualisation, R.F.-R.; methodology, M.U.K., M.S. and R.F.-R.; formal analysis, M.U.K. and M.S.; investigation, M.U.K., M.S. and R.F.-R.; data interpretation: M.U.K., M.S. and R.F.-R.; resources, R.F.-R. and R.G.; data collection, N.H., C.J. and R.F.-R.; writing—original draft preparation, M.U.K. and M.S.; writing—review and editing, M.G., G.C., R.G. and R.F.-R.; supervision, R.F.-R. and R.G.; project administration, R.F.-R.; funding acquisition, R.F.-R. All authors have read and agreed to the published version of the manuscript.

**Funding:** This research received no external funding.

**Institutional Review Board Statement:** The experimental procedures involving human subjects described in this paper were approved by the University of Canberra's Human Ethics Committee (Number: 11837).

**Informed Consent Statement:** Informed consent was obtained from all subjects involved in the study.

**Data Availability Statement:** The data that support the findings of this study are available from R.F.R., upon reasonable request.

**Conflicts of Interest:** The authors declare that they have no known competing financial interests or personal relationships that could have appeared to influence the work reported in this paper.

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
