# Peer review of "Multilevel Pain Assessment with Functional Near-Infrared Spectroscopy: Evaluating ΔHBO2 and ΔHHB Measures for Comprehensive Analysis"

_sensors, doi:10.3390/s24020458_

Round 1

Reviewer 1 Report

Comments and Suggestions for Authors

The manuscript targets an important problem of assessing pain using functional near infrared spectroscopy (fNIRS) with application to evaluating pain in non-verbal patients. Data is collected from 30 healthy individuals.  Pain was induced using transcutaneous electrical nerve stimulation. Pain tolerance and pain threshold were considered as high pain and low pain, respectively. fNIRS data (24 channels) was collected concurrently from the prefrontal cortex and analyzed. Classifiers were trained to detect three classes of no pain, low pain and high pain. Results are presented and discussed.

I have the following comment.

Major comments:

1) For pain experiments, more details are needed to clarify how the experiments were done during pain-inducing sessions. For example, if the pain tolerance (used to indicate high pain) is noted as the “highest intensity of pain the participant could endure before reaching a point of intolerable discomfort”, how the individuals were able to tolerate this level for six 10-s sessions? In Figures 3 and 4, it appears that high pain sessions on arm and hand were done consecutively, which brings up the concern of the tolerance of the individuals. A description of how individuals tolerated these sessions would be necessary. Inclusion of a figure illustrating the data collection paradigm (similar to the work of authors in [21]) would be helpful.

2) The authors use a simple relative range operator to identify channels with high amplitudes. It is not clear why this method is beneficial compared to other commonly-used “bad-channel” detection methods. Additionally, it is not clear for what duration of the acquired signals RR is calculated. There is also a typo in the denominator of equation (1).  Authors should mention what were the maximum and minimum number of remained channels for subjects after this process. Was there a large variations?

3) The authors grouped the data corresponding to inducing high pain to arm and high pain to hand into one class, and low pain to arm and hand in another class for the purpose of classification. This requires assuming that the impacts of pain on arm and hand in terms of activating the prefrontal cortex, are similar. Please provide justification for this assumption. Indeed, based on Figure 7, the impact of pain on hand and arm appears to be actually different. Can authors show results of pain-level classification for arm and hand separately? 

4) Additionally, the authors indicate that they included “six additional observations from the rest period of each subject, prior to pain stimulation” as the baseline class. Were these collected prior to or after low pain sessions or high pain sessions? Please clarify.

5) The deoxy-Hb signal is generally considered as the weak signal in fNIRS studies compared to oxy-Hb. Figure 7 further confirms this. It is surprising to see similar accuracy results obtained in Tables 3 and 4 when using features from oxy-Hb and dexoy-Hb separately. Can authors provide explanation for this?

6) The inclusion of features from both dexoy-Hb and oxy-Hb has slightly improved the classification results compared to using each signal separately. However, the number of used features is almost doubled. This issue needs to be discussed. That is, what would be the justification for using a larger number of features and increasing the computational complexity for the purpose of gaining slight improvement in the outcome?

7) Similarly, the inclusion of feature selection based on MRMR have slightly improved the performance or reduced only few features in some cases (comparing Tables 3 and 4). This brings up the question of its necessity, and it requires further discussions.

Minor comments:

1)  HbO2 and HHb are not “modalities” of fNIRS, they represent oxygenated and deoxygenated Hb. fNIRS measures the changes in the “concentration” of HbO2 and HHb. To be accurate, please do not refer to these as “modalities” of fNIRs. Also, use Delta[HbO2] and  Delta[HHb] to indicate changes (Delta) in the concentration ([ ]) when referring to these throughout the paper and in the figures/tables.

2) Furthermore, this reviewer believes it is inaccurate to call the model that uses information from oxy-Hb and deoxy-Hb signals, a “multi-modal” approach.  The information from both signals is acquired from the same neuroimaging modality.  Both signals have the same temporal/spatial resolution and both represent changes in the blood oxygenation levels. Calling a model using information from these two signals a multi-modal model is inaccurate.

3) Literature review should be more comprehensive and needs to include the recent growing work on fNIRS and pain using machine learning.

4) Figure captions need to be more detailed. For example, do Figures 3 and 4 reflect the data from the same subject? How many channels are shown in Figure 4? What does the color bar represent in Figure 7? Are the results shown in Figure 7 averaged across subjects or they represent results from one subject? etc.

Reviewer 2 Report

Comments and Suggestions for Authors

This is a study analyzed multiple fNIRS modalities including not only HBOs but also HHB in to investigate comprehensive evaluation of pain levels, highlighting the SVM’s effectiveness in pain assessment. According to their analysis combined with machine learning techniques, the authors concluded HBOs and HHB would be potential biomarkers for high intensity pain and for low intensity pain respectively.

This study shows interesting results that lead further studies subjected to patients with chronic pain.

Please find below my comments:

Major comments

1.        Please define why the authors used HHB besides HBOs as fNIRS parameters with corresponding references. Generally, HBOs is used for fNIRS because of its reliability and uncertainty of HHB. A part of it was mentioned in discussion section; but when it was in introduction section, the interesting results of HHB in the present study will be more meaningful.

2.        Is there any difference between stimulation on hand and arm? Why they were analyzed together categorized as only low/high pain groups, but not separately hand/arm-low/high pain groups?

3.        The conclusion section is quite verbose. Please summarize main points and keep in compact with a few sentences.

Minor comment

1.        Please define what HBOs and HHB indicate at the first use in abstract.

Reviewer 3 Report

Comments and Suggestions for Authors

After completion of the assessment work on this revised manuscript entitled on 'Multilevel Pain Assessment with Functional Near-Infrared Spectroscopy: Evaluating HBO2 and HHB Modalities for Comprehensive Analysis', in current form it is unsuitable to recommend for publication due to poor language.

Never use 'we' as well as 'our' type pronouns in scientific article. Remove such pronouns and improve language.

Revise abstract with respect to sentences and make it concise, precise for ease of understanding.

Hence, minor revision is essential in order to make it suitable for further thorough reviewer work.

Comments on the Quality of English Language

Poor quality of the english langualge.

Never use 'we' as well as 'our' type pronouns in scientific article. Remove such pronouns and improve language.

Author Response

We thank the reviewer for the comments. Although we believe that using "we" or "our" helps us convey a sense of personal involvement in the research process, we have reviewed and refined the language used, and are pleased to present the revised content in its improved form.